# The Brightest Point in Accretion Disk and Black Hole Spin: Implication to the Image of Black Hole M87*

**Vyacheslav I. Dokuchaev [1,2,]*** and **Natalia O. Nazarova [3]**

1    Institute for Nuclear Research of the Russian Academy of Sciences, Moscow 117312, Russia
2    Moscow Institute of Physics and Technology, 9 Institutskiy per., Dolgoprudny, Moscow 141700, Russia
3    Scuola Internazionale Superiore di Studi Avanzati (SISSA), Via Bonomea 265, 34136 Trieste (TS), Italy
*    Correspondence: dokuchaev@inr.ac.ru

**Abstract:** We propose the simple new method for extracting the value of the black hole spin from the direct high-resolution image of black hole by using a thin accretion disk model. In this model, the observed dark region on the first image of the supermassive black hole in the galaxy M87, obtained by the Event Horizon Telescope, is a silhouette of the black hole event horizon. The outline of this silhouette is the equator of the event horizon sphere. The dark silhouette of the black hole event horizon is placed within the expected position of the black hole shadow, which is not revealed on the first image. We calculated numerically the relation between the observed position of the black hole silhouette and the brightest point in the thin accretion disk, depending on the black hole spin. From this relation, we derive the spin of the supermassive black hole M87*, $a = 0.75 \pm 0.15$.

**Keywords:** general relativity; black holes; black hole shadow

## 1. Introduction

The Event Horizon Telescope (EHT) recently presented for the first time the impressive direct image of the supermassive black hole in the galaxy M87 [1–6]. There are two specific features on this image: the dark silhouette and the bright region.

It is obvious to suppose that the bright region is related to the emission of accreting matter around the supermassive black hole M87*. However, what is the physical origin of the dark silhouette, viewed on the presented image? It depends on the black hole astrophysical environment.

The black hole shadow is viewed by the distant observer in the case of a stationary luminous background, placed beyond the unstable circular photon orbit $r_{\rm ph}$ [7–22]. In the pioneering work of J. M. Bardeen [7], the black hole shadow was called the "apparent boundary" of the black hole. The outline of the black hole shadow is shown by the dark red closed curves in our figures.

The quite different case is the black hole image produced by the luminous non-stationary matter plunging into black hole in a region between the unstable circular photon orbit (photon ring) $r_{\rm ph}$ and the black hole event horizon $r_{\rm ph}$. The evident example of the luminous non-stationary matter is the inner region of thin accretion disk between the Innermost Stable Circular Orbit $r_{\rm ISCO}$ (see definitions and details, e.g., in [23]) and the event horizon $r_{\rm h}$. A non-stationary luminous matter in this region is spiraling down toward the event horizon and produces a dark image of the event horizon silhouette instead of the black hole shadow [8,24–31]. In addition, the outline (contour) of this silhouette is the image of the event horizon equator [32–34].

We calculate below the form of the event horizon silhouette of the rotating Kerr black hole by modeling the emission of the non-stationary luminous matter spiraling down into black hole in the

inner region of thin accretion disk adjoining the event horizon. The trajectories of photons, reaching the distant observer from the inner region of the thin accretion disk, are calculated in the geometric optic approximation. It is also supposed that the accretion disk is optically thin. The event horizon silhouette is smaller than the black hole shadow and is placed on the celestial sphere within the awaited position of black hole shadow.

We identify the dark region on the EHT image with the event horizon silhouette and, respectively, the bright region on the EHT image with the brightest point in the accretion disk. For the case of the supermassive black hole M87*, we find the angular distance of the brightest point in the thin accretion disk from the center of the observed event horizon silhouette depending on the black hole spin parameter $a$. As a result, from this dependence, we find the spin of the supermassive black hole M87*, $a = 0.75 \pm 0.15$.

## 2. Photon Trajectories in the Kerr Metric

We describe the trajectories of photons (null geodesics) in the Kerr metric by using the standard Boyer–Lindquist coordinate system [35] with coordinates $(t, r, \theta, \varphi)$ and with units $G = c = 1$. Additionally, we put $mG/c^2 = 1$. With these units, the dimensionless radius of the black hole event horizon is $r_h = 1 + \sqrt{1 - a^2}$, where the dimensionless spin parameter of the Kerr black hole is $0 \le a \le 1$.

Trajectories of particles with a rest mass $\mu$ in the Kerr space-time are determined by three constants of motion: a total energy $E$, a component of angular momentum parallel to symmetry axis (azimuth angular momentum) $L$ and the Carter constant $Q$, which is related to the non-equatorial motion of particles [9,36]. Particle trajectories (geodesics) in the radial and latitudinal directions are defined, correspondingly, by a radial effective potential

$$R(r) = [E(r^2 + a^2) - La]^2 - (r^2 - 2r + a^2)[\mu^2 r^2 + (L - aE)^2 + Q] \tag{1}$$

and a latitudinal effective potential

$$\Theta(\theta) = Q - \cos^2\theta[a^2(\mu^2 - E^2) + L^2/\sin^2\theta]. \tag{2}$$

Trajectories of photons (null geodesics with $\mu = 0$) in the Kerr space-time are determined only by two dimensionless parameters, $\lambda = L/E$ and $q^2 = Q/E^2$. These parameters are related to the impact parameters on the celestial sphere $\alpha$ and $\beta$ seen by the distant observer placed at a given radius $r_0 >> r_h$ (i.e., practically at infinity), at a given latitude $\theta_0$ and at a given azimuth $\varphi_0$ (see, e.g., [7,37] for more details):

$$\alpha = -\frac{\lambda}{\sin\theta_0}, \quad \beta = \pm\sqrt{\Theta(\theta_0)}, \tag{3}$$

where $\Theta(\theta)$ is from Equation (2).

We use integral equations of motion for photons [9,36] for numerical calculations of the gravitational lensing by the Kerr black hole

$$\fint^r \frac{dr}{\sqrt{R(r)}} = \fint^\theta \frac{d\theta}{\sqrt{\Theta(\theta)}}, \tag{4}$$

$$\varphi = \fint^r \frac{a(r^2 + a^2 - \lambda a)}{(r^2 - 2r + a^2)\sqrt{R(r)}} dr + \fint^\theta \frac{\lambda - a\sin^2\theta}{\sin^2\theta\sqrt{\Theta(\theta)}} d\theta, \tag{5}$$

$$t = \fint^r \frac{(r^2 + a^2)P}{(r^2 - 2r + a^2)\sqrt{R(r)}} dr + \fint^\theta \frac{(L - aE\sin^2\theta)a}{\sqrt{\Theta(\theta)}} d\theta, \tag{6}$$

where the effective potentials $R(r)$ and $\Theta(\theta)$ are from Equations (1) and (2). The integrals in Equations (4), and (5) are understood to be path integrals along the trajectory.

The path integrals in (4), (5) and (6) are the ordinary ones for photon trajectories without the turning points. For example, the path integral integral Equation (4) can be written in this case through the ordinary integrals as

$$\int_{r_s}^{r_0} \frac{dr}{\sqrt{R(r)}} = \int_{\theta_s}^{\theta_0} \frac{d\theta}{\sqrt{\Theta(\theta)}}. \tag{7}$$

In the case of photon trajectories with one turning point $\theta_{\min}(\lambda, q)$ (an extremum of latitudinal potential $\Theta(\theta)$), integral Equation (4) can be written through the ordinary integrals as

$$\int_{r_s}^{r_0} \frac{dr}{\sqrt{R(r)}} = \int_{\theta_{\min}}^{\theta_s} \frac{d\theta}{\sqrt{\Theta(\theta)}} + \int_{\theta_{\min}}^{\theta_0} \frac{d\theta}{\sqrt{\Theta(\theta)}}. \tag{8}$$

In general, a lensed black hole produces an infinite number of images [37–40]. With the exception of the very special orientation cases, the most luminous image is a so-called direct or prime image, produced by photons that do not intersect the black hole equatorial plane on the way to a distant observer. In the meantime, the secondary images (named also like the higher order images or light echoes) are produced by photons that intersect the black hole equatorial plane several times. The energy flux from secondary images as a rule is very small in comparison to one from the direct image.

## 3. Black Hole Shadow

A black hole shadow is the gravitational capture cross-section of photons from the stationary luminous background placed at a radial distance from a black hole exceeding the radius of unstable photon circular orbit $r_{\mathrm{ph}}$ (see the definition in [23]).

A black hole shadow in the Kerr metric, projected on the celestial sphere and seen by a distant observer in the equatorial plane of the black hole, is determined from the simultaneous solution of equations $R(r) = 0$ and $[rR(r)]' = 0$, where the effective radial potential $R(r)$ is from Equation (17). The corresponding solution for the black hole shadow (for a distant observer in the black hole equatorial plane) in the parametric form $(\lambda, q) = (\lambda(r), q(r))$ is

$$\lambda = \frac{-r^3 + 3r^2 - a^2(r+1)}{a(r-1)}, \quad q^2 = \frac{r^3[4a^2 - r(r-3)^2]}{a^2(r-1)^2} \tag{9}$$

(see, e.g., [7,9] for more details).

Quite a different black hole image is produced in the case of a black hole highlighted by the non-stationary luminous matter plunging into a black hole inside the radius of unstable photon circular orbit $r_{\mathrm{ph}}$.

## 4. Event Horizon Silhouette

The observed dark event horizon silhouette is recovered by gravitational lensing of photons emitted in the innermost part of the accretion disk adjoining to the event horizon. In a geometrically thin accretion disk, placed in the equatorial plane of the black hole, there is an inner boundary for stable circular motion, named the marginally stable radius or the Inner Stable Circular Orbit (ISCO), $r = r_{\mathrm{ISCO}}$ (see, e.g., [23] for more details):

$$r_{\mathrm{ISCO}} = 3 + Z_2 - \sqrt{(3 - Z_1)(3 + Z_1 + 2Z_2)}, \tag{10}$$

where

$$Z_1 = 1 + (1 - a^2)^{1/3}[(1+a)^{1/3} + (1-a)^{1/3}], \quad Z_2 = \sqrt{3a^2 + Z_1^2}. \tag{11}$$

The corresponding values of parameters $E$ and $L$ for particles in the accretion disk co-rotating with the black hole at a circular orbit with a radius $r$ are obtained from the simultaneous solution of equations $R = 0$ and $dR/dr = 0$, where the effective radial potential $R$ is from Equation (1):

$$\frac{E}{\mu} = \frac{r^{3/2} - 2r^{1/2} + a}{r^{3/4}(r^{3/2} - 3r^{1/2} + 2a)^{1/2}}, \tag{12}$$

$$\frac{L}{\mu} = \frac{r^2 - 2ar^{1/2} + a^2}{r^{3/4}(r^{3/2} - 3r^{1/2} + 2a)^{1/2}}. \tag{13}$$

We use a simple model for describing the non-stationary motion of the small gas elements of accreting matter in the region $r_{\mathrm{h}} \le r \le r_{\mathrm{ISCO}}$ and suppose the pure geodesic motion of the separate small gas element (or compact gas clump) in the accretion flow with the conserved orbital parameters $E$ and $L$ from (12) and (13), corresponding to the radius $r = r_{\mathrm{ISCO}}$. See in Figures 1 and 2 the examples of numerically calculated 2$D$ trajectories of small disc elements spiraling down into the black holes in the region $r_{\mathrm{h}} \le r \le r_{\mathrm{ISCO}}$.

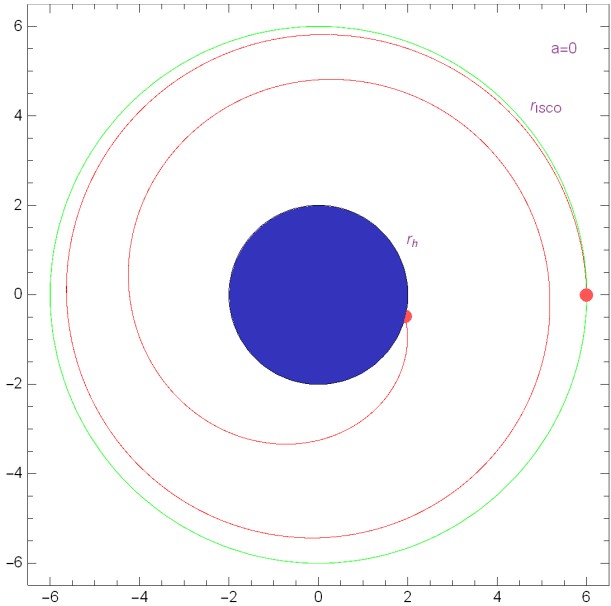

**Figure 1.** 2$D$ trajectory (red spiral) of the small accretion disk element in the non-stationary region $r_{\mathrm{h}} \le r \le r_{\mathrm{ISCO}}$, spiraling down into the non-rotating Schwarzschild black hole with $a = 0$. The small accretion disk element is starting at $r = r_{\mathrm{ISCO}} = 6$ (green ring) with orbital parameters $E/\mu = E(r_{\mathrm{ISCO}})/\mu = 2\sqrt{2}/3$ and $L/\mu = L(r_{\mathrm{ISCO}})/\mu - 0.001 = \sqrt{3}/2 - 0.001$, where $E$ and $L$ are from (12) and (13).

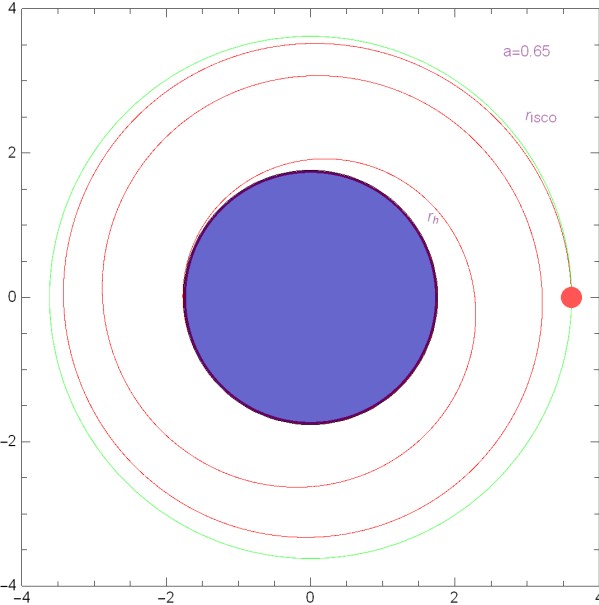

**Figure 2.** 2*D* trajectory (red spiral) of the small accretion disk element in the non-stationary region $r_{\rm h} \leq r \leq r_{\rm ISCO}$, spiraling down into the rotating black hole with $a = 0.65$. The small accretion disk element is starting at $r = r_{\rm ISCO} = 3.616$ (green ring) with orbital parameters $E/\mu = E(r_{\rm ISCO})/\mu = 0.903$ and $L/\mu = L(r_{\rm ISCO})/\mu - 0.001 = 2.675 - 0.001$, where $E$ and $L$ are from (12) and (13). In contrast with the non-rotating black hole, the spiraling down accretion disk element is multiply winding around the rotating black hole by approaching the event horizon at $r = r_{\rm h} = 1.759$.

We suppose also that the thin accretion disk is transparent and the energy flux in the rest frame of small gas elements is isotropic and conserved during their spiraling down into the black hole.

See in Figure 3 the parameters of photon trajectories $\lambda$ and $q$, reaching the distant observer at $r_0 \gg r_{\rm h}$ from the rings in thin accretion disk in the black hole equatorial plane at $\theta_s = \pi/2$ and $r_s = 0.01 r_{\rm h}$ and $r_s = r_{\rm ISCO}$. The pairs $(\lambda, q)$ are the numerically calculated solutions of integral Equations (4) and (8). Namely, the pairs of solutions $(\lambda, q)$ of Equation (4), corresponding to the photon trajectories without the turning points, are shown by the blue colors. Respectively, the pairs of solutions $(\lambda, q)$ of Equation (8), corresponding to the photon trajectories with one turning point in latitudinal direction at $\theta = \theta_{\rm min}(\lambda, q)$, are shown by the red colors. The case of the rotating black hole M87* with spin $a = 0.75$ is shown. It is physically reasonable to suggest that the black hole spin axis is aligned with the large-scale jet. In the case of the supermassive black hole M87*, the line of sight makes a $\theta_0 = 163°$ angle with the spin axis of the black hole. This value of the inclination angle is derived from the detailed VLBI (Very Long Baseline Interferometry) observations of the famous relativistic jet from the M87* [41]. Additionally, these VLBI observations indicate the clockwise flow rotation about the jet axis.

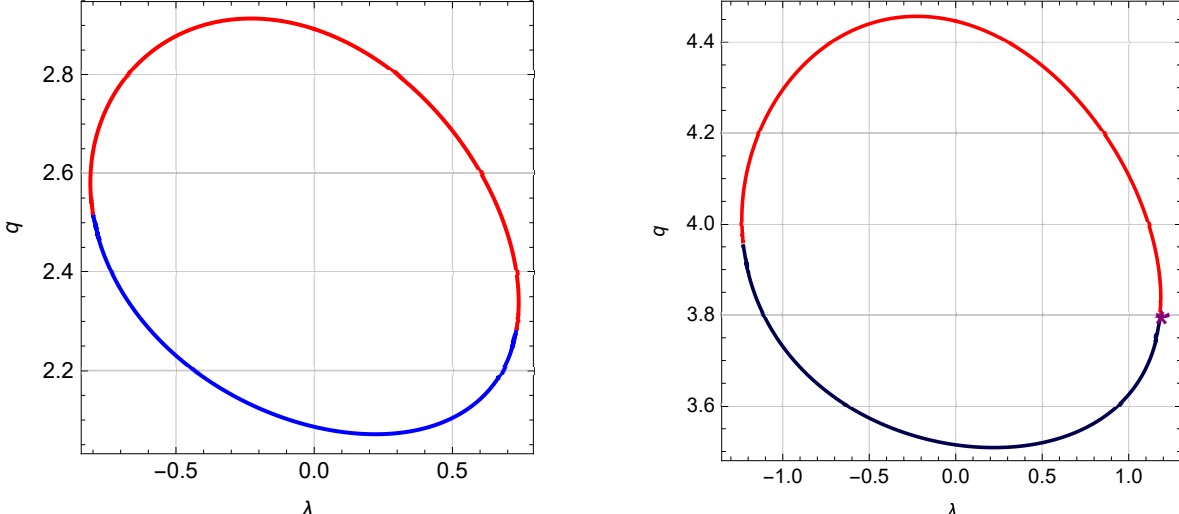

**Figure 3.** Parameters of photon trajectories $\lambda$ and $q$, reaching the distant observer from the rings in accretion disk at $r = 1.01r_{\mathrm{h}}$ (**left graph**) and at $r = r_{\mathrm{ISCO}}$ (**right graph**). The case of rotating black hole with the spin $a = 0.75$ in the galaxy M87 is shown, corresponding to $\theta_0 = 163°$. The blue color corresponds to photon trajectories without the turning points, defined from numerical solutions of the integral Equation (7). The red color corresponds to photon trajectories with only one turning point at $\theta = \theta_{\min}(\lambda, q)$, defined from numerical solutions of the integral Equation (8). The brightest point (marked by a star "★" at the right graph) in the accretion disk is placed at radius $r_{\mathrm{ISCO}} \simeq 1.16r_{\mathrm{h}}$ and corresponds to the photon trajectory without turning points and with the maximum permissible azimuth angular momentum. Parameters of photon trajectory from the brightest point are $\lambda = 1.18$ and $q = 3.79$ or $\alpha = -4.03$ and $\beta = -0.18$.

To calculate the energy shift of photons and energy flux from the lensed image of accretion disk, it is necessary to take into account both a red-shift in the black hole gravitational field and Doppler effect. It is convenient to use in these calculations the orthonormal Locally Non-Rotating Frames (LNRF) [23,42], for which the observers' world lines are $r = const$, $\theta = const$, $\varphi = \omega t + const$, where a frame dragging angular velocity

$$\omega = \frac{2ar}{(r^2 + a^2)^2 - a^2(r^2 - 2r + a^2)\sin^2\theta}. \tag{14}$$

The requested photon energy in the frame, comoving with the small disk element (or compact gas cloud) at $\theta = \pi/2$, is [34,38]

$$\mathcal{E}(\lambda, q) = \frac{p^{(t)} - V^{(\varphi)}p^{(\varphi)} - V^{(r)}p^{(r)}}{\sqrt{1 - [V^{(r)}]^2 - [V^{(\varphi)}]^2}}. \tag{15}$$

In this equation, the azimuth velocity $V^{(\varphi)}$ of the small disk element with orbital parameters $E$, $L$ and $Q = 0$ relative the LNRF, falling in the equatorial plane onto a black hole, is

$$V^{(\varphi)} = \frac{r\sqrt{r^2 - 2r + a^2}\,L}{[r^3 + a^2(r+2)]E - 2aL}. \tag{16}$$

A corresponding radial velocity in the equatorial plane relative the LNRF is

$$V^{(r)} = -\sqrt{\frac{r^3 + a^2(r+2)}{r}}\,\frac{\sqrt{R(r)}}{[r^3 + a^2(r+2)]E - 2aL}, \tag{17}$$

where $R(r)$ is defined in (1) with $Q = 0$. The components of photon 4-momentum in the LNRF are

$$p^{(\varphi)} = \sqrt{\frac{r}{r^3 + a^2(r+2)}}\, \lambda, \; p^{(t)} = \sqrt{\frac{r^3 + a^2(r+2)}{r(r^2 - 2r + a^2)}}\, (1 - \omega\lambda), \tag{18}$$

$$p^{(r)} = -\frac{1}{r}\sqrt{\frac{(r^2 + a^2 - a\lambda)^2}{r^2 - 2r + a^2} - [(a-\lambda)^2 + q^2]}. \tag{19}$$

Respectively, the energy shift (the ratio of photon frequency at infinity to a corresponding one in the rest frame of the small disk element is $g(\lambda, q) = 1/\mathcal{E}(\lambda, q)$. This energy shift is used in numerical calculations of the energy flux from the accretion disk elements measured by the distant observer, following formalism by C. T. Cunnungham and J. M. Bardeen [37]. The corresponding values of the observed local energy flux from the non-stationary region $r_{\text{ISCO}} \leq r \leq r_{\text{h}}$ in the accretion disk are marked in Figures 4–6 by the different levels of color intensity: a more intensive color corresponds to a more intensive energy flux. In the meantime, the different colors from the dark blue to red in these figures correspond to the red-shifts of observed photons. Near the brightest points in these figures, the blue color of observed photons is chosen.

The form of the black hole silhouette does not depend on the observed frequency and is defined only by the properties of the black hole gravitational field. For this reason, we are free to choose any frequency (or color) for emission from the brightest point in the accretion disk. We choose the dark blue color for the brightest point, viewed by the distant observer. The observed frequencies (colors) from the other points in the non-stationary region $r_{\text{ISCO}} \leq r \leq r_{\text{h}}$ are scaled (normalized) relative to the chosen frequency at the brightest point in accordance with the calculated energy shift of the observed photons from Equation (15).

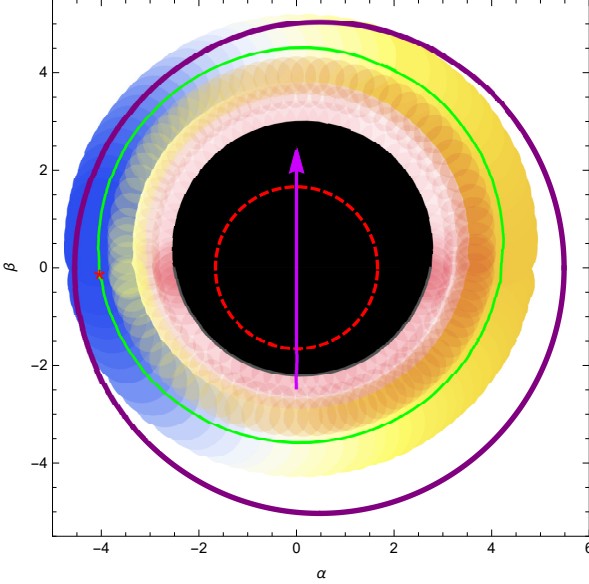

**Figure 4.** The internal part of the lensed accretion disk at $r_{\text{h}} \leq r \leq r_{\text{ISCO}}$ adjoining to the event horizon in the case of rotating black hole with spin $a = 0.75$. The brightest point in accretion disk is at radius $r_{\text{ISCO}} = 3.158$ (green ring) and corresponds to the photon trajectory with $\lambda = 1.179$, $q = 3.788$, 1.179, 3.788 $\alpha = -4.029$, $\beta = -0.18$. The black region is the event horizon silhouette. The viewed black silhouette in the case of M87* is the image of the southern hemisphere of the black hole horizon. The outline (contour) of this black silhouette is the equator of the event horizon. The closed dark red curve is a border of the black hole shadow. The dashed red circle is the observed position of the black hole event horizon in the imaginary Euclidian space. The magenta arrow is the black hole rotation axes.

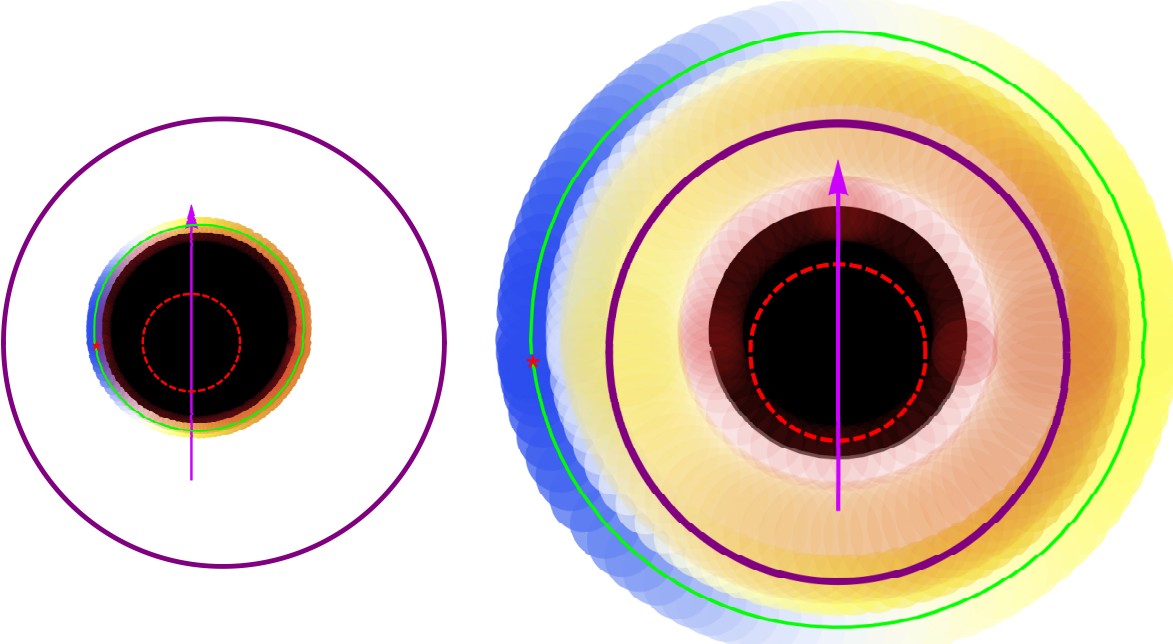

**Figure 5.** Silhouette of the southern hemisphere of the black hole event horizon (black region) projected inside the black hole shadow (closed purple curves) for the black hole in the galaxy M87, $\theta_0 = 163°$. Positions of the brightest points in accretion disk at $r = r_{\text{ISCO}}$ (green ring) are marked by red stars ⋆ in the case of black hole spin $a = 0.9882$ (left image) and $a = 0$ (right image), respectively.

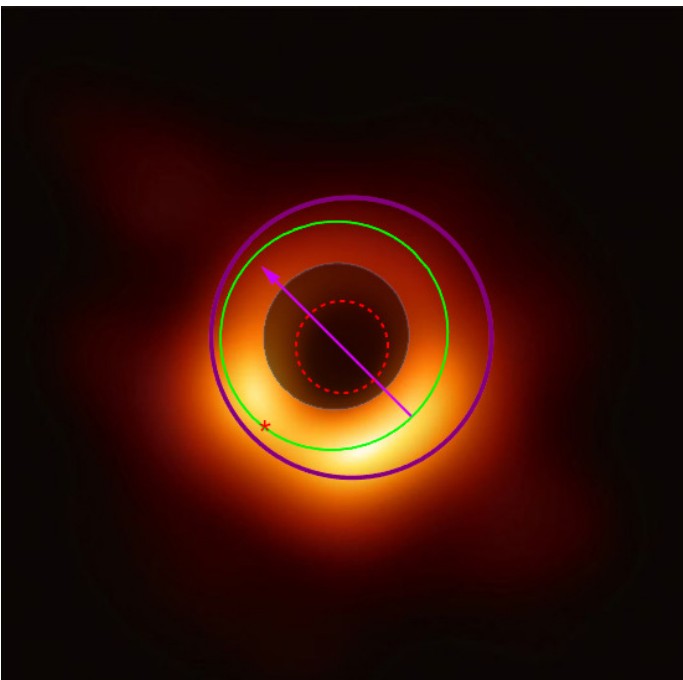

**Figure 6.** Superposition of the M87* image, obtained by EHT [1–6], and the thin accretion disk model from Figure 5 in the case of black hole spin $a = 0.75$. Star ⋆ here and as well as in the all other similar figures marks the modeled position of the brightest point in the thin accretion disk at $r = r_{\text{ISCO}}$ (green ring).

## 5. Spin of the Black Hole M87*

Direct images of the inner non-stationary part of thin accretion disk in the region $r_h \leq r \leq r_{ISCO}$, adjoining to the black hole event horizon, are presented in Figures 2, 3, 5 and 7 for the case of M87*. In these Figures the black region is the event horizon silhouette. The viewed black silhouette in the case of M87* is the image of the southern hemisphere of the black hole horizon. The outline (contour) of this black silhouette is the equator of the event horizon. The closed dark red curve is a border of the black hole shadow. The dashed red circle is the observed position of the black hole event horizon in the imaginary Euclidian space. The magenta arrow is the black hole rotation axes.

The gravitational red-shift and Doppler effect are taken into account in these images. Local artificial colors of the thin accretion disk images are related with an effective local black-body temperature of the accreting gas elements. We made numerical calculations for 10 different values of the black hole spin in the range $0 \leq a \leq 1$ with the step 0.1. All these numerical calculations demonstrate that the observed brightest point in the thin accretion disk is always placed at radius $r = r_{ISCO}$ at the point corresponding to photon trajectory without turning point and with the maximum permissible azimuth angular momentum $\lambda > 0$. In Figure 3, the position of the brightest point is marked by a star "★" at the right graph, corresponding to the largest (positive) azimuth angular momentum $\lambda$ of photon with the direct orbit, starting from $r = r_{ISCO}$ and reaching the distant observer without the turning points.

The derived relation between the position of the brightest point in the thin accretion disk and the observed black hole silhouette depending on the black hole spin is shown in Figure 8. From the comparison of this relation with the image obtained by the EHT, it follows that the best fit for the spin of the supermassive black hole of M87* is $a = 0.75 \pm 0.15$. In this fit, the value $6 \ 10^9 M_\odot$ for the mass of the supermassive black hole of M87* [43] is used. The $1\sigma$ error in this fit corresponds to the observed width of the asymmetric bright ring on the M87* image [1–6]. The derived value of the M87* spin is in a general agreement with the other similar estimations [44–52]. See in Figure 5 the superposition of the M87* image and the modeled image of the thin accretion disk in the case of $a = 0.75$.

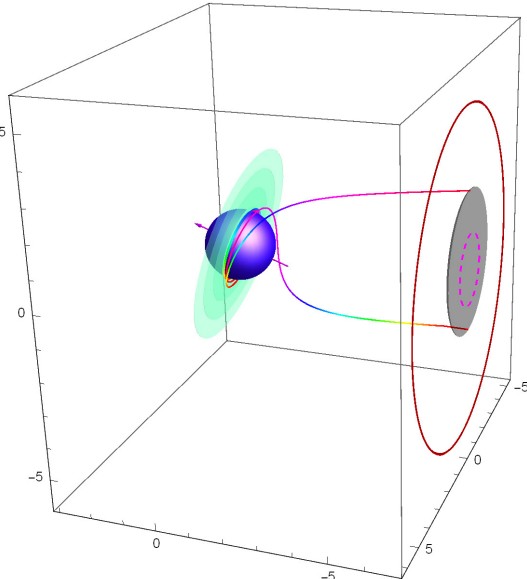

**Figure 7.** Two 3D photon trajectories starting from the thin accretion disk (light green oval) at $r = 1.01 r_h$ in the equatorial plane of the black hole with $a = 0.9982$ and reaching the distant observer very near the outline (contour) of the event horizon silhouette (light black region). In the case of M87*, the viewed dark silhouette is the image of the southern hemisphere of the black hole horizon. Parameters of these photon trajectories are $\lambda = -0.047$ and $q = 2.19$ and, respectively, $\lambda = -0.029$, $q = 1.52$. The closed dark red curve is the corresponding outline of the black shadow, defined by Equation (9). The dashed red circle is the observed position of the black hole event horizon in the imaginary Euclidian space. The magenta arrow is the black hole rotation axes.

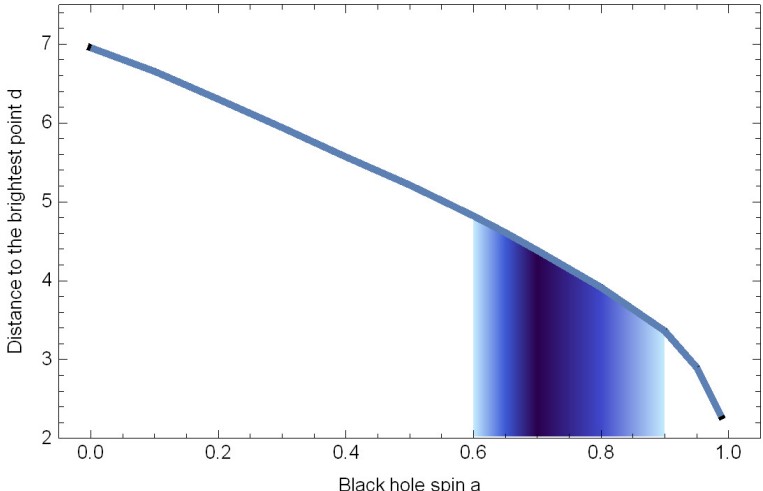

**Figure 8.** The numerically calculated relation between the distance $d(a)$ of the brightest point in the thin accretion disk from the center of the viewed black hole silhouette depending on the black hole spin $a$ for the case of M87*.

## 6. Conclusions

We calculated numerically the form of the event horizon silhouette of the rotating Kerr black hole by modeling the emission of non-stationary luminous matter spiraling down into the black hole in the inner region of thin accretion disk at $r_{\text{ISCO}} \leq r \leq r_{\text{h}}$. It was supposed (i) that the motion of the separate small gas element (or compact gas clump) in the accretion flow is purely geodesic; (ii) that the thin accretion disk is transparent and (iii) that the energy flux in the rest frame of small gas elements is isotropic and conserved during their spiraling down into the black hole. The resulting form of the event horizon silhouette does not depend on the local emission of accretion disk and governed completely by the gravitational field of the black hole. In the case of the supermassive black hole M87*, viewed at the inclination angle 163°, a dark silhouette on the EHT image is the southern hemisphere of the black hole event horizon. The contour of this silhouette is the equator of the event horizon.

The brightest point in accretion disk corresponds to the largest (positive) azimuth angular momentum $\lambda$ of photon with the direct orbit, starting from $r = r_{\text{ISCO}}$ and reaching the distant observer without the turning points. By using the first image of the black hole M87*, we find the value of the black hole spin $a = 0.75 \pm 0.15$.

It is clearly seen in the Figures 4, 5 and 6 that the effective horizontal and vertical diameters of the black hole shadow for the case of M87*, with the black hole rotation axis oriented nearly opposite to the direction toward the distant observer, very weakly depend on the black hole spin. These diameters are approximately 2 times greater than the corresponding ones for the dark silhouette of the black hole event horizon viewed on the first image of the supermassive black hole M87*. This means that the large awaited size of the black hole shadow is not reconciled with the size of the dark black region on the first black image presented by the EHT.

**Author Contributions:** Investigation, V.I.D. and N.O.N. The authors contributed to the paper equally.

**Funding:** This research was funded by Russian Foundation for Basic Research grant 18-52-15001 NCNIa.

**Acknowledgments:** We are grateful to E.O. Babichev, V.A. Berezin, Yu. N. Eroshenko and A.L. Smirnov for stimulating discussions.

**Conflicts of Interest:** The authors declare no conflict of interest.

## Abbreviations

The following abbreviations are used in this manuscript:

EHT　　Event Horizon Telescope
LNRF　Locally Non-Rotating Frames
VLBI　Very Long Baseline Interferometry

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
