# Peer review of "The Brightest Point in Accretion Disk and Black Hole Spin: Implication to the Image of Black Hole M87*"

_universe, doi:10.3390/universe5080183_

Round 1

Reviewer 1 Report

After a rather general and well-known introduction about geodesics in Kerr spacetimes, and after overlaying some photon trajectories with the M87* black hole image, the spin value of a=0.75 is magically obtained for this black hole.

Even if not many details are given, the overall analysis done in his paper is overly simplistic, and one which I do not believe fully captures the complex Physics involved here. I therefore cannot recommend that this article be published.

Reviewer 2 Report

This manuscript is investigated black hole shadow image of M87* obtained by EHT to constraint the black hole spin parameter by their analytical approach. Original analytical work has done in separated paper by authors (Dokuchaev, Nazarova, & Smirnov 2019, GRG, 51, 81) and in this manuscript, the authors are applied it to M87*. I think this kind of research is important for the understanding the physics of black hole shadow but the author’s approach is too simple to apply M87* case and obtained results are misreading. Therefore I can not recommend this manuscript to be published in Universe at this current version.

The authors should be addressed following my issues.

Major comments:

1.     The authors consider stationary thin accretion disk and the non-stationary luminous matter spiraling down into black hole. But in this manuscript, there is no information about the matter distribution of both disk and accretion flow. How the matter is accreted onto black hole from accretion disk? What kind of mechanism do you consider for angular momentum transport? The authors should write the physical condition of accreting matter and thin accretion disk.

2.     Recently Gralla et al. (2019) have also considered the emission from matter near the black hole (between the event horizon and accretion disk). I think this is very similar approach for this manuscript. What is the difference? The authors should be addressed and explained the difference from their work.

3.     M87* is categorized low luminous AGN and mass accretion rate is far below the Eddington ratio. In this situation, accretion flow onto a black hole is described by the radiation inefficient accretion flow (RIAF) or advection dominated accretion flow (ADAF) which is optically thin and geometrically thick hot accretion flow. Therefore the author’s considered thin accretion disk (standard disk or Shakura Sunyaev disk) is not relevant for M87*. The authors should be reconsidered to using thin disk model in M87*.

4.     About the event horizon silhouette, which frequency do you consider seeing it? In EHT observation, they used 230GHz radio emission. The authors should use this frequency for the comparison. I think emitted photon near the event horizon is highly redshifted and it is not seen at 230GHz image (it is not much bright).

5.     What radiation process do you considered? What emissivities and absorptivities are used? The authors should explain more detail of your physical assumption.

6.     For Figs 2 to 5, the authors do not explain these figures in the text (looks just cited). The authors should extend the explanation of these figures. Otherwise the reader does not understand.

7.    The authors estimated black hole spin value from the comparison with your analytical work. What is the black hole spin direction? How do you decide it? What is the flow rotation direction? Why negative spin case (a<0) is ruled out?

8.     We know M87 has a strong jet flow in large scale. The authors should give a comment about the formation of jet which expect to launch near the black hole and relationship to shadow image.

Minor comments:

1.     In line 22, r_{ph} should be r_h.

2.     In line 44, Eqs (17) and (2), I think it is Eqs. (1) and (2).

3.     In Figure 1, the authors mentioned r=0.01r_h. Do you consider to set inside event horizon?

4.     In Figure 3, what is the color scale means? What is the different lines? What is the pink arrow? The authors should write more explanation of figure.

Reviewer 3 Report

Report:

Manuscript ID universe-538521, entitled "Brightest Point in Accretion Disk and Black Hole Spin:

Implication to the Image of Black Hole M87*" submitted to Universe

The aim of the authors is

to shown that  

a dark region in the recent image of the Black hole M87* is consistent with a silhouette of the black hole event horizon. 

They calculate the a dependence between the brightens point in an accretion disk with the black hole silhouette. 

The authors used some results about the separability on the geodesics equations on the Kerr space-time to solve the trajectories of photons that reach the observer and infer a value for the spin parameter and the view angle. 

There are however some aspects in their calculations that are not clear for the reader. Here I mention some of them

The authors solve the integral equation (7) but they do not specify that is the variable to solve nor the parameters in the limits of integration. 

The authors claim that the value of the spin parameter is a = 0.75 \pm 0.15

How do they estimate and error for the spin of the black hole? 

In the paragraph after Eq. 6 (and in the equation itself) there are symbols that are not defined. e.g. $P$, $V_r$, $V$

The explanation about the results of Figure 1 (lines 68-70) is not clear to me. I recommend the authors a rephrasing  

In Fig. 2, How is the projection on the box frame taken?

Labels on the box frame may help the reader to understand the figure. 

It is not clear to me  when the authors state "This energy shift is used in numerical calculations

 of the energy flux from the accretion disk ..." Do they use the same formalism in their calculations? or they simply inform that the formalism can be used?

In Line 80, where the authors say:

"numerical calculations demonstrate ..." it is not clear at all, how do they support this affirmation. Could the authors give more detail on this issue?

Could the authors explain what was the model of emission used in their sentence:

"We calculated numerically a form of the event horizon silhouette of the rotating Kerr black hole by

modelling the emission of non-stationary luminous matter spiralling"

In summary, recommend the authors give more details on the construction of the disks and the models they used.

Taking into account these issues, I consider the manuscript is not suitable for publication in the present form and a major revision is needed. 

Round 2

Reviewer 2 Report

In the revised manuscript, the authors revised it extensively and addressed some of my raised issues. However, I still have several issues which I could not satisfy the author’s comments. Therefore I can not recommend this manuscript to be published in Universe at this current version.

The authors should be addressed and commented following my issues:

1.     The authors consider the accreting matter from r < r_ISCO which is spiralling down onto black hole. Therefore considering thin disk is not much important in this manuscript. It would be fine but I still not fully understand the situation of accreting matter onto black hole. How do you determine the angular momentum of accreting matter?  Does it give arbitrary? If the accreting matter is coming from thin accretion disk located at r_ISCO, the accreting matter should have the rotational velocity (angular momentum) determined by thin accretion disk solution (Shakura-Sunyaev model). The authors should write more detail situation in the manuscript.

2.     We could not fully agree with the author’s opinion about ADAF model on M87*. Observationally ADAF model is supported in M87. M87 is low-luminous AGN. Such low luminosity is related lower mass accretion rate and the physical solution of accretion flow structure in low mass accretion rate is advection dominated accretion flow (ADAF, e.g., Narayan & Yi 1994, 1995; Reynolds et al. 1996). ADAF model can reproduce the broad-band spectra (SED) observed in M87 (e.g., Yuan & Narayan 2014, Prieto et al. 2016). In SED of M87, we could not see the blue bump feature, nor is the Fe K \alpha line which are feature of existence of thin accretion disk. I’m fine the authors considered thin disk model in this manuscript. But the authors should write the weakness of thin accretion disk assumption in the manuscript.

3.     The authors are still not clearly provided the observed frequency in this manuscript. In EHT observation has taken at 230GHz, 1.3 mm radio wavelength. If the authors want to compare with EHT results, the authors should show the shadow images in the same frequency. Otherwise the comparison is non-sense.

4.     Even horizon silhouette is interesting. But authors also said the emission is highly redshifted near the horizon. I still not understand why we can observe the emission from them.  If we consider at 230GHz frequency and proper radiation process from the accreting matter, such emission is very weak in comparison with photon ring. The authors should provide the information clearly.

5.     Related point 4, I still not understand why photon ring emission from the radiation near the photon orbit is weaker than event horizon silhouette. I think in EHT observed shadow image, photon ring emission is dominated component of bright ring structure.

Reviewer 3 Report

The authors have improved the manuscript and most of the issues I pointed were solved.

Thus I recommend the manuscript for its publication.

Author Response

Point 1: Comments and Suggestions for Authors. The authors have improved the manuscript and most of the issues I pointed were solved. Thus I recommend the manuscript for its publication.

Response 1: Thanks for the positive decision after reviewing.

Round 3

Reviewer 2 Report

The authors answered my comments and improved the manuscript. I do not have any further comments about the manuscript. Therefore I recommend that this manuscript should be published in MDPI journal Universe.